# The Riboflavin Metabolism in Four *Saccharomyces cerevisiae* Wine Strains: Assessment in Oenological Condition and Potential Implications with the Light-Struck Taste

**DOI:** 10.3390/jof9010078

**Published:** 2023-01-05

**Authors:** Alessandra Di Canito, Alessio Altomare, Daniela Fracassetti, Natalia Messina, Antonio Tirelli, Roberto Foschino, Ileana Vigentini

**Affiliations:** 1Department of Food, Environmental and Nutritional Sciences (DeFENS), Università degli Studi di Milano, Via G. Celoria 2, 20133 Milan, Italy; 2Department of Biomedical, Surgical and Dental Sciences (DSBCO), Università degli Studi di Milano, Via della Commenda 10, 20122 Milan, Italy

**Keywords:** flavin derivatives, methionine, metabolism, yeasts, *S. cerevisiae*, wine, light-struck taste

## Abstract

Riboflavin (RF), or vitamin B2, is an essential compound for yeast growth and a precursor of the flavin coenzymes, flavin mononucleotide (FMN) and flavin adenine dinucleotide (FAD), involved in redox and non-redox processes. RF is a photosensitive compound involved in the light-struck taste (LST), a fault causing the formation of off-flavors that can develop when the wine is exposed to light in the presence of methionine (Met), as well. As both RF and Met can be associated with detrimental changes in wines, a better comprehension of its yeast-mediated production is relevant to predict the maintenance of the desired character of the wine. This study aims at assessing the production of flavin derivatives (FDs) and Met by *S. cerevisiae* oenological starters under laboratory conditions. The results showed the presence of extra- and intracellular FDs, and Met is a strain-dependent characteristic being also affected by the initial content of RF in the medium. This finding was confirmed when the winemaking was carried out in a relevant environment. Our results evidenced the important impact of the yeast strain on the content of RF and its derivatives.

## 1. Introduction

Riboflavin (RF) is an essential vitamin for yeast growth and a precursor of flavin coenzymes, flavin mononucleotide (FMN) and flavin adenine dinucleotide (FAD). Despite in just-made grapes, only negligible RF content was detected (a few tens of µg/L). Its concentration in wine can reach an average concentration of around 200 μg/L and even up to about 320 µg/L due to the metabolism of yeasts, mainly *Saccharomyces cerevisiae*, releasing RF during the alcoholic fermentation (AF) [1,2,3].

RF is one of the main actors involved in the occurrence of the light-struck taste (LST), a sensory fault described as cooked cabbage or vegetables also leading to the loss of fruity and floral notes. This defect can appear when the white or rosé wines bottled in clear glass are exposed to light, especially in the UV-Vis spectra (370–450 nm) due to oxidative reactions [1,4,5,6,7,8,9]. At these wavelengths, light-sensitive molecules in wine, photosensitizers, can reach a high energy state inducing physical and chemical changes [1,10]. Specifically, RF is a photosensitive compound that, if exposed to light, generates a strongly unstable triplet state, and its low energy state is obtained when RF is fully reduced. This reduction can occur when RF acquires two electrons from electron-donor compounds [11]. In particular, methionine (Met), present in wine at 3–4 mg/L on average [12,13], is able to donate the two electrons for the full reduction of RF. The reaction between RF and Met leads to the formation of methional, which rapidly converts to methanethiol (MeSH), whose condensation of two molecules forms dimethyl disulfide (DMDS) [14], responsible for the off-flavor perception [15,16,17]. Moreover, the formation of DMDS could also originate from a dimer cation radical species, a reaction occurring in a short time and without oxidizing species [16]. When RF is lower than 50–80 µg/L, the risk of LST occurrence decreases [2,11,18].

Although oenological strategies aimed at reducing the RF content in wine at the end of AF (use of fining agents, phenolic compounds, and specific packaging) have been developed [6], the responsibility of yeast in the production of RF is still poorly investigated. Actually, RF and its active forms (FMN and FAD) serve as co-factors in enzymatic reactions of several central metabolic pathways, thus they are fundamental for yeast survival [19]. The RF accumulation in *S. cerevisiae* has been suggested to be a strain-dependent character in grape must fermentations [1,18], thus supporting the hypothesis that different genetic backgrounds can distinguish the yeast strains for their capability in modulating the RF pathway. Indeed, RF and flavin nucleotide biosynthetic pathways have been deeply investigated in *S. cerevisiae* under laboratory conditions [20,21,22,23,24,25,26]. After all, molecular experiments to obtain *S. cerevisiae* deletion mutants of the genes involved in RF biosynthetic pathway (*RIB1*, *RIB2*, *RIB3*, *RIB4*, *RIB5*, *RIB7*) has been reported; these studies demonstrated the essential nature of the FDs because the generated mutants became non-viable.

In addition, the possibility of cells obtaining RF from the extracellular environment has been examined in *S. cerevisiae* [27,28], and the plasma membrane transporter Mch5p has been identified as the main protein in charge of the RF uptake from the growth medium [29,30]. However, the mechanisms involved in the utilization of exogenous RF in the pathway or the transporter regulation are still unknown.

Similarly, Met plays a crucial role in yeast survival being involved in fundamental metabolisms, such as protein synthesis and cell signaling [29].

Even though the RF metabolism in yeasts is highly studied, specifically for biotechnological interests, a few articles underline the relationship between the metabolism of RF and Met, and the oenological conditions ([1] only for RF, [18]). As far as we know, no study specifically shows in detail the flavin compounds and Met productions by *S. cerevisiae* during the process to obtain sparkling wines, which is produced by two fermentations steps both responsible of the RF accumulation.

Since metabolic pathways can be strongly influenced by nutrient availability, the present study aimed at assessing how the presence/absence of RF in the growth medium can affect the metabolism of RF and its derivatives in *S. cerevisiae* wine strains. In particular, we focused our attention on both laboratory and oenological conditions. As it is not possible to identify a strain unable to produce these two compounds, in applicative terms understanding if the medium of growth influences the metabolisms of RF and Met may help in developing the knowhow to manage their accumulation in wines.

The question was intended to establish if the presence of RF at the beginning of the fermentation could serve as a signal to report a sufficient vitamin availability to cells. At this purpose, the intra- and extracellular contents of RF, its related co-factors and Met were monitored during the AF of four commercial starter yeasts growing in synthetic media (similar in composition to a grape must and containing 0 or about 125 µg/L of RF to test the absence or high content) and Chardonnay must. The used laboratory cultural media were formulated in order to support the yeast growth in the tumultuous fermentative phase in terms of nutrients with or without RF. This experimental plan allowed to compare the results obtained in the synthetic media with the ones from a real grape must, where the content of RF is usually too low for triggering LST [13]. Moreover, a vinification trial in a relevant environment was performed to assess the reliability of the results in a large-scale experiment.

## 2. Materials and Methods

### 2.1. Yeast Strains and Cultural Media

Four commercial *S. cerevisiae* strains were used in this work: Lalvin EC1118 (Lallemand Inc., Blagnac cedex, France), IOC18-2007 (Lallemand Inc., Blagnac cedex, France), Fermivin LS2 (Oenobrands SAS, Montpellier, France) and AWRI796 (Maurivin, Kuils River, South Africa). Yeasts were maintained at −80 °C in YPD medium (10 g/L yeast extract, 20 g/L peptone, 20 g/L glucose) supplemented with 20% (*v/v*) glycerol.

Two laboratory growth media were prepared on the base of synthetic grape must composition (SM; glucose 115 g/L, fructose 115 g/L, tartaric acid 5 g/L, malic acid 3 g/L, citric acid 0.2 g/L, l-arginine 2 g/L, pH = 3.5) [30,31], modified as follow: (i) SMV, not containing RF, supplemented with essential vitamins and salts for yeast growth as described in Verduyn et al. 1992 [32]; (ii) SMV containing 125 µg/L RF (SMV+RF). Concentrated stock solutions of each additional compound were prepared, filtered through 0.22 μm nitrocellulose membranes (Millipore filter, type GSWP, Billerica, MA, USA) and added in adequate amounts. Chardonnay must (2014 harvest; sugars 213 g/L, titratable acidity (TA) 8.8 g tartaric acid/L, pH 3.28, yeast assimilable nitrogen (YAN) 168 mg N/L [1,11]), containing 30.5 µg/L of RF, was clarified by centrifugation at 4 °C at 8000× *g* × 20 min (Avanti J25, Beckman, CA, USA) and was added of sodium metabisulfite (Na_2_S_2_O_5_) at the final concentration of 30 mg/L.

### 2.2. Yeast Inoculum, Growth Conditions, and Fermentations

Yeast pre-cultures of the four strains were prepared in YNB (Difco™, ThermoFischer Scientific, Waltham, MA, USA) added with 20 g/L glucose as carbon source and incubated at 25 °C, 120 rpm (SSL1 Orbital Shaker, Stuart Scientific, Staffordshire, U.K.). After 72 h, cells were collected by centrifugation at 2876× *g*, 15 min (Hettich, ROTINA 380R, Tuttlingen, Germany), washed twice with sterile distilled water and used to inoculate the synthetic grape must medium and the Chardonnay must at the concentration of 1–5 × 10^6^ CFU/mL, respectively. Experiments were performed in 250 mL flasks, filled with 100 mL aliquots, closed with Muller valves (containing 10% (*v/v*) HCl) to obtain oxygen-limited conditions, and incubated at 25 °C ± 2 °C in static conditions. To avoid any influence on the yeast growth by the sample withdrawing, two flasks were prepared per each strain and cultural media, analyzed at 10 and 25 days and discarded. Fermentations were carried out in triplicate.

For each point, including the time at the inoculum (0 days), measurement of OD_660nm_ and the analysis of CFU/mL were carried out, plating the sample on YPD medium added with agar 18 g/L (Scharlau, Barcelona, Spain). Finally, supernatants and cells (corresponding to 25 OD_660nm_ units per sample) were collected and stored at −20 °C, in absence of light to avoid the photooxidation of the samples, for further analyses.

### 2.3. Preparation of Cell Extracts

Cells were subjected to a lysis protocol to obtain the extracts for the evaluation of intracellular content of RF derivatives. First, a volume of cell culture corresponding to 25 OD_660nm_ units per sample was centrifuged at 2876× *g* × 15 min (Hettich, Tuttlingen, Germany). The pellet was suspended in a solution containing 0.9 M sorbitol, 0.1 M EDTA, 14 mM β-mercaptoethanol and 500 µg/mL Zymoliase 100T (USBiological, Salem, MA, USA) and incubated at 30 °C overnight. The suspension was centrifuged at 2876× *g* × 15 min (Hettich, Tuttlingen, Germany) to remove the supernatant; then, the obtained spheroplasts were resuspended in 1 mL of sterile distilled water, added with an iso-volume of 0.45 mm glass beads (Sigma-Aldrich, St. Louis, MO, USA) and broken by vortexing for 10 min. The mixture was incubated at 25 °C overnight. Then, the suspension was incubated at 65 °C for 30 min, and after at 4 °C for the same time of incubation. Finally, the sample was centrifuged at 23,000× *g* × 10 min (Hettich, MIKRO 200, Tuttlingen, Germany) and the supernatant was collected. The lysis was verified by microscope observation (Standard 25, Zeiss, Oberkochen, Germany) counting the remaining whole cells by Thoma cell counting chamber (Graticules Optics Limited, Cambridge, U.K.) to evaluate the rate of lysed cells, in terms of “number of unbroken cells/numbers of total cells) × 100”; the standard error of the procedure was calculated to be <3%.

### 2.4. Vinification Trials in a Relevant Environment

A large-scale experiment was performed in order to assess the reliability of the results obtained in the laboratory-scale experiments. Triplicate trials were conducted with Chardonnay must (50 L) in vintage 2022. The must was analyzed for sugar contents, pH, total acidity, and YAN by an ISO 9000-accredited laboratory (Enoconsulting, Erbusco, BS, Italy) using Fourier transform infrared spectroscopy. The experiments were performed using EC1118 and AWRI796 strains, inoculating the active dry yeasts following the producer’s instructions. Then, the content of RF, flavin compounds, and Met was assessed in the must, at the start and at the end of the fermentations as described below (Section 2.6).

### 2.5. Fermentation Monitoring

To monitor the yeast growth, viable cells were enumerated by plate counts (CFU/mL). Moreover, the fermentative process was followed quantifying the amount of residual sugars (glucose and fructose), the acetic acid, glycerol and ethanol productions using enzymatic determinations associated with the automatic enzymatic analyzer iCUBIO i-Magic M9 (r-Biopharm, Darmstadt, Germany) according to the instructions of the manufacturer. The enzymatic kits are specific for the investigated substances; no interferences have been observed during the experiments. When necessary, appropriate dilutions of the samples were prepared in order to obtain a linear measurement of the tested compounds.

### 2.6. Determination of Flavin Derivatives and Methionine

FDs and Met, in both the supernatants and the cell extracts, were quantitatively measured by ultra-performance liquid chromatography (UPLC), as reported in Fracassetti et al. 2018, 2019, 2021 [6,11,33], with slight modifications.

RF, FAD, and FMN were determined by UPLC (Acquity system, Waters, Milford, MA, USA) equipped with a fluorescence detector Acquity UPLC^®^ (Waters) using a Hypersil ODS C18, 3 µm, 100 × 3 mm (CPS Analitica, Milan, Italy). The flow rate was 0.5 mL/min, and the injection volume was 10 µL. The samples were filtered using a 0.22 μm PVDF filter prior to the injection. The elution solvents were citrate buffer 50 mmol at pH 2.5 (solvent A) and methanol (solvent B) in gradient mode, in which B was from 5% to 100% in 13.50 min, followed by column washing and equilibration. The quantification was carried out by the external method with a nine-point calibration curve in concentration that ranged from 1 µg/L to 500 µg/L (R^2^ = 0.997–0.999). The detection was performed at 420 nm and 530 nm for excitation and emission, respectively [11]. Data acquisition and processing were performed by Empower 3 software (Waters).

In order to normalize the production of RF and related co-factors per biomass level (ng/cell), the intracellular volumetric concentrations (µg/L) of RF, FMN, and FAD were divided per 1 mL (volume in which the cellular samples were resuspended) and then divided again per 25 OD_660nm_. As far as the extracellular production, the volumetric concentrations (µg/L) of RF, FMN, and FAD were divided for the biomass (OD_660nm_/mL) at the corresponding sampling time. All data, intracellular and extracellular, were expressed as ng/10^6^ cells by taking into account that 1 OD_660nm_ corresponds to 10^6^ cells. Note that the concentration of RF, FMN, and FAD (µg/L) possibly contained in the cultural media prior to the inoculum was subtracted from the detected concentrations at 10 and 25 days.

Finally, the total potential amount of RF molecules produced by cells was calculated also considering that the RF skeleton represents the chemical block for the construction of flavone compounds (FMN and FAD). Thus, an estimation at a specific sampling time can be obtained by summing intracellular and extracellular volumetric contents of RF molecules plus the content of two coenzymes (by subtracting the initial amount and dividing them for their molecular weight). The number of potential molecules of RF were obtained by multiplying the found number of moles by the Avogadro’s number. The obtained results were further simplified to 10^13^ order of magnitude.

Met concentrations were quantified by UPLC as *o*-phthalaldehyde (OPA) derivatives under the conditions described by Fracassetti et al. 2019 and 2021 [6,11] with some modifications. The derivatization solution was prepared in a 10 mL volumetric flask by dissolving 250 mg of OPA in 1.5 mL of ethanol, adding 200 μL of 2-mercaptoethanol, and making up to the volume with borate buffer 0.4 M at pH 10.5. The pre-column derivatization was performed as follows: 500 μL of borate buffer 0.4 M at pH 10.5 were added with 50 μL of sample and 100 μL of OPA solution; the reaction mixture was vortexed for 5 min, filtered with 0.22 μm PVDF filers (Millipore) and injected. The chromatographic separation of OPA derivatives was carried out using an Acquity UPLC (Waters) system equipped with a fluorescence detector Acquity UPCL^®^ (Water). The separation was carried out with a Kinetex, 5 μm EVO C18, 100 A, 150 × 2.1 mm (Phenomenex, Torrance, CA, USA) maintained at 40 °C. The flow rate was 1 mL/min and the injection volume was 10 μL. The elution solvents were (solvent A) citrate buffer 10 mM at pH 7.5 and (solvent B) methanol in gradient mode in which B was from 5% to 47% in 22 min. The quantification was carried out with the external method by using a six-points calibration curve at concentration in the range 0.5–20 mg/L. The detection was performed at 335 nm and 440 nm for excitation and emission, respectively. Data acquisition and processing were performed by Empower 2 software (Waters). Met was determined in the synthetic media and in must, and at day 25.

### 2.7. Statistical Analysis

Statistical analysis was performed with SPSS Win 12.0 program (SPSS Inc., Chicago, IL, USA). Factorial ANOVA was carried out to identify the significant differences within yeast strains, sampling times, and media. Significant differences were judged by the post hoc Fischer LSD test (*p* < 0.05). The principal component analysis (PCA) was performed with Statistica 12 software (Statsoft Inc., Tulsa, OK, USA) on auto-scaled data for an overall overview of the release of RF, FMN, FAD, and Met due to the different investigated yeast strains, cultural conditions, and sampling points.

## 3. Results

This section presents the results of the investigation of FDs and Met production by *S. cerevisiae* wine strains during the fermentations carried out in three different growth media: two synthetic grape musts, that differ each other for the presence of RF (namely SMV and SMV+RF) and a Chardonnay must (containing a minor amount of RF compared to SMV+RF). These cultural media were selected to investigate how the RF release or uptake from the environment occurs in yeast cells in oenological conditions. The production of the flavin compounds was evaluated in both oenological and biological perspectives, considering the volumetric release and the potential RF molecular production, respectively. The fermentations were analyzed after 10 and 25 days, besides the starting point. The mid time point was selected to carry out a physiological comparison among the growth of the strains in the three media, since preliminary results revealed that yeasts maintained a high cell viability at day 10th in comparison to the end of alcoholic fermentation (residual sugars < 10 g/L) (data not shown). The 25th day after the starter’s inoculum was also considered in the experimental plan as the final point of fermentation to obtain data compatible with a real vinification process. Finally, the results of the vinification trials prove the reliability of the results obtained in the laboratory-scale experiments.

### 3.1. Yeast Growth and Fermentative Performance

Growth trends of the four selected *S. cerevisiae* strains are shown in Figure 1. In general, the results revealed no significant differences among mean counts of the strains (*p* = 0.95) at each time, while a significant difference related to the growth media (*p* < 0.001) was observed. For the four strains, an average count of approximately 7 ± 0.2 Log (CFU/mL) and 6 ± 0.3 Log (CFU/mL) was detected, as an average for the four strains, after 10 days of fermentation in SMV and SMV+RF media, respectively. Then, at the final point (25 days), the yeast population decreased at about 1 ± 0.1 Log (CFU/mL) in both synthetic media. In contrast, the yeast cell viability remained almost constant during the fermentation process in Chardonnay must, starting from a value of biomass equal to 6 ± 0.1 Log (CFU/mL) to reach the maximal population of 7 ± 0.1 Log (CFU/mL) after 10 days, and to finally return to almost 6 ± 0.1 Log (CFU/mL) at the end of the process.

The analysis of the chemical parameters associated with the fermentation included acetic acid, glycerol, and ethanol at the three sampling times. Acetic acid production in both SM-based media ranged between 0.6 and 2.0 g/L at 25 days, with AWRI796 strain showing the highest accumulation with respect to the other yeasts regardless of the growth media (*p* < 0.001) (Figure 2). Acetic acid was mainly released after 10 days by all the strains in both synthetic media, while the lowest amount of was found in Chardonnay must (around 0.3 g/L at 10 days for all the strains), revealing significant differences among the mean values obtained in the diverse type of growth media (*p* < 0.001). Glycerol content ranged between 5 and 7 g/L, 4 and 6 g/L, and 6 and 6.5 g/L on the 10th day for SMV, SMV+RF, and Chardonnay must, respectively (Figure 2). Even if these results showed parallel trends among the strains, significant differences among the growth environment were revealed (*p* < 0.001). Similarly, comparable ethanol productions (*p* > 0.05) were found when the yeast strains grew in the same medium, whereas significant differences were shown among the mean values observed in SMV, SMV+RF, and must (*p* < 0.001) (Figure 2).

### 3.2. Determination of Extracellular Flavin Derivatives

The release of RF, FMN, and FAD was assessed for the four investigated strains at all the cultural conditions adopted (Figure 3). A significant increase in RF and its co-factors was found with a kinetic dependence upon the growth media and the strain.

During the fermentation in SMV (the flavin-free medium) the FDs were produced by all the strains investigated (Figure 3A). In particular, at day 10th, RF was released at 30.4 ± 3.7 µg/L by EC1118, 34 ± 0.1 µg/L by IOC18-2007, 32.4 ± 1.5 µg/L by LS2 and 52.2 ± 1.4 µg/L by AWRI796. These results showed the major ability of AWRI796 strain in releasing RF (+62% in comparison to the mean of other strains). After 25 days, a decrease up to 6–8 µg/L of the RF content in the supernatant was detected for all the other strains. As observed for the RF, FMN and FAD were secreted in the cultural media, reaching average values of about 5.3-folds and 6.6-folds higher than RF, respectively, for FMN and FAD. On the 10th day EC1118, IOC18-2007 and LS2 released a higher amount of FAD (255–269 µg/L) than of FMN (192–230 µg/L), which were partially consumed after 25 days. Contrarily, AWRI796 released the lowest amount of FMN and FAD (−20% and −24%, respectively, as the average quantities discharged by the other yeast). FMN excretion showed a great variability among the strains at the 10th day and then on the 25th day the values were settled, while for AWRI796 no differences were observed among the mid and the final times of sampling. In addition, for FAD release, a significant difference in AWRI796 after 10 and 25 days was noted with respect to the other strains.

Similar to what was observed in SMV, RF increased for all the strains after 10 days in SMV+RF medium (Figure 3B) that already contained RF (125.8 ± 0.6 µg/L). In comparison to the other three strains, AWRI796 released the significantly highest amount of RF at 10 days (+31% of the main value of the others), which is 2-fold higher in comparison to the SMV medium. On the 25th day, a decreasing accumulation trend was seen. Furthermore, AWRI796 showed a major release of FMN after 10 days (71 ± 3.7 µg/L), proving again to be different from the other strains. Only EC1118 significantly increased the FMN release on the 25th day. Regarding the concentration of FAD in the medium, AWRI796 released the lowest amount of this compound on the 10th day and at the end of the fermentation (102.3 ± 15.5 µg/L) in comparison to the other strains (−22%).

In the Chardonnay must, at starting time, the FDs were detected up to 30.5 ± 0.0 µg/L, 35.6 ± 0.0 µg/L and 11.1 ± 0.0 µg/L for RF, FMN, and FAD, respectively. During the fermentation, the RF content in the supernatant was comparable after 10 days for the investigated strains, whereas after 25 days, significant differences were found, with the AWRI796 strain releasing the most (Figure 3C). In particular, after 10 days, LS2 and AWRI796 strains released the highest amount of RF (74 ± 5.6 µg/L), but the other two strains produced a similar concentration (66.7 ± 1.5 µg/L for IOC18-2007 and 71 ± 2.7 µg/L for EC1118). RF concentration further increased at 25th day reaching the amount of: 103.3 ± 4.1 µg/L for EC1118, 94.8 ± 2.5 µg/L for IOC18-2007, 95.1 ± 4.1 µg/L and 114.5 ± 3.5 µg/L for AWRI796. The concentration of FMN in must was constant for all the investigated strains (9 ± 0.5–10 ± 0.1 µg/L). The only differences were observed for EC1118 and AWRI796 strains during their growth between the 10th and the 25th days. Finally, the concentrations of FAD significantly increased on the 25th day for all the strains, with the exception of LS2. At the end of the fermentation, FAD was 58.6 ± 5.5 µg/L, 61.4 ± 7.6 µg/L, 46.5 ± 7.7 µg/L and 63.7 ± 10.1 µg/L for EC1118, IOC18-2007, LS2 and AWRI796, respectively.

In order to understand the influence of yeast strains and growth conditions on the release of RF and its co-factors, the principal component analysis (PCA) was carried out (Figure 4). Four significant principal components explain 83% of the variance. In particular, the first two components explained 37% and 17% of the variance, respectively, for P1 and P2. The yeast strains can be grouped into two clusters: the first cluster includes the strains EC1118, IOC18-2007, and LS2, while the second contains the AWRI796 strain exclusively. This result suggests that the AWRI796 strain has a completely different behavior compared to the other strains under investigation. Moreover, the release of RF resulted in the opposite of FAD and FMN, while the two RF derivatives depend on each other. Considering the influence of growth media, the release of FAD and FMN was related to SMV medium, while that of RF to SMV+RF medium (Figure 4).

### 3.3. Determination of Intracellular Flavin Derivatives

Results of the analysis of the intracellular content of RF and its co-factors are reported in Table 1. As regards the yeasts growth in the SMV medium, no significant difference among the strains was observed for RF, as values remained stable between 10 and 25 days (6.3–11.3 µg/L). FMN was detected only after 25 days in EC1118 cells, while similar intracellular FAD quantities among the strains were revealed during the fermentation but not detected at 25 days in the extract of AWRI796.

On the other hand, the RF content of the cell extracts increased in cells grown in SMV+RF medium from the 10th to 25th day, except for EC1118 and LS2 strains (+19% for IOC18-2007 and +41% for AWRI796). Moreover, the RF content of AWRI796 cells was significantly higher than the other strains on the 25th day of fermentation. Instead, it was not possible to investigate differences in the FMN content because it was not detected in all intracellular samples. Concerning FAD content, a decrease was observed during the fermentation from 10 to the last day for all the strains except for IOC18-2007, as follows: −52% for EC1118, −100% for LS2, and −10% for AWRI796.

A different scenario was observed when the cells were grown in Chardonnay must, revealing great variability among the strains and the fermentation times. RF intracellular content after 10 days was higher than that recorded in the synthetic media: 45.5 ± 7.8 µg/L for EC1118, 62.1 ± 0.04 µg/L for IOC18-2007, 52.5 ± 6.6 µg/L for LS2, and 77.5 ± 10.1 µg/L for AWRI796; then, a relevant decrease in intracellular RF, up to 35%, was observed after 25 days of fermentation. Similar concentrations of RF were determined on the 25th day for all the strains. Regarding the intracellular FMN, it was uniquely detected after 10 days in all the cell extracts but not after the last day of fermentation. Moreover, on the 10th day of fermentation, higher concentrations of intracellular FAD were observed, revealing significant differences among the strains. Finally, after 25 days, a reduction of 12% and 8.6%, respectively, for IOC18-2007 and AWRI796, was assessed, while in the cell extracts of the other two strains, FAD was not detected.

The PCA showed three principal components explaining 64% of the variance (Figure 5). In particular, the P1 explains 36% of the variance. The yeast strains can be clustered in two groups, including the strains EC1118, LS2, and AWRI796, while the IOC18-2007 strain differs in the content of intracellular FDs.

### 3.4. Potential Production of Molecular RF

The potential production of molecular RF was calculated in order to compare the physiological behavior of the four strains. The highest amount of RF molecules was produced by the four strains during their growth in an SMV medium (Table 2). Indeed, in this condition, the total quantity of RF was in the range of 430–559 ng/10^6^ cells. While during the fermentation in SMV+RF, the values of the produced RF molecules were almost halved for all the strains except for AWRI796, for which a lower amount of −25% and −35%, considering the 10th and the 25th day, respectively, was observed. On the contrary, the lowest amount of RF molecules resulted in Chardonnay must; indeed, after 10 days of fermentation, the produced molecules were in a range between 70 and 80 ng/10^6^ cells, and then they increased until 138, 120, 108.3, and 162 ng/10^6^ cells, respectively for EC1118, IOC18-2007, LS2, and AWRI796.

### 3.5. Determination of Extracellular Methionine

Also, the release of Met at the end of fermentation was considered due to its influence on the formation of the off-flavors associated with the occurrence of LST [6]. Obtained data from the chemical analysis were reported in Table 3. The initial amount of Met was comparable in the two synthetic grape must media (9.7 ± 1.1 and 9.5 ± 1.1 mg/L, respectively, for SMV and SMV+RF), while it was slightly higher in the must (10.8 ± 1.1 mg/L). However, significant differences were observed in the release of this amino acid on the 25th day. In SMV, EC1118, and IOC18-2007 extracellular content of Met remained stable, while a little increase was found for the other strains (+1.9 mg/L and +2.6 mg/L, respectively, for LS2 and AWRI796).

Conversely, in the SMV+RF medium, Met concentration increased for all the strains. Particularly, Met release was significantly different only for AWRI796 compared to EC1118 and LS2, releasing +2.8 mg/L at the end of the fermentation compared to +4–5 mg/L of the other strains.

In Chardonnay must, the strains consumed the most of the Met content after 25 days, with an average decrease of around 30% during the fermentation processes. Moreover, in this condition, the behavior of the strains was comparable since the only significant differences were highlighted comparing the starting and the final points of the fermentation process.

Moreover, PCA showed that two significant principal components explain 42% of the variance (Figure 6). The four yeast strains can be grouped into two clusters, including EC1118, IOC18-2007, and LS2 strains in the first cluster and AWRI796 strains in the second one.

### 3.6. Vinification Trials in a Relevant Environment

Chardonnay must be used for the vinification trials was characterized by 192.4 g/L of sugars, 8.0 g tartaric acid/L (TA), 288 mg N/L (YAN), and pH of 3.09. The content of RF and its derivatives was determined as 8.17 ± 2.2 µg/L, 2.32 ± 1.0 µg/L, and 1.96 ± 0.56 µg/L for RF, FMN, and FAD, respectively (Figure 7). In terms of the difference in the concentration of the analyzed compounds from the start to the end of the fermentation processes, a great variation was revealed for RF, while FMN and FAD delta values resulted in comparable. Indeed, in the winery trials, an RF increase of about 30% for EC1118 (corresponding to 105.1 µg/L vs. 72.8 µg/L in laboratory experiments) and 49% for AWRI796 (corresponding to 167.0 µg/L vs. 84.0 µg/L in laboratory experiments) was detected. Finally, the Met extracellular content was detected just in traces in all the fermentation trials (data not shown).

## 4. Discussion

Since yeast cell life and resilience to stress factors can be strongly influenced by nutrient availability, clarifying which growth conditions interfere with the metabolism of RF can help to limit its release during grape must fermentation. In this work, the effect of exogenous RF on the FDs biosynthesis has been evaluated in synthetic media and real Chardonnay musts; particularly, we were going to determine if extracellular RF content might serve as a signal to report sufficient vitamin availability to cells. Moreover, the outcome of the study could improve the required knowledge for the setup of suitable approaches to counteract RF accumulation in wine. For this purpose, the used cultural media (similar to real musts) were formulated in order to support the yeast growth in the tumultuous fermentative phase in terms of nutrients, except for the presence of RF: SMV was lacking in the vitamin content, while SMV+RF contained about 125 µg/L RF. The experiments in Chardonnay must have been included in the experimental plan, though RF concentration, in this case, was usually too low for triggering LST [13].

Plate counts confirmed that the viability remained stable on the 10th day (primary fermentation phase), and then it started to decline slightly [34,35]. Moreover, the analysis of the fermentative metabolites allowed us to confirm that in all the tested conditions, the four selected strains were able to complete a simulated fermentation process, confirming the results described by Zhang and co-authors [36].

The biosynthetic pathways of RF and its derivative coenzymes have been deeply investigated in *S. cerevisiae*; they are strictly related since FMN, and FAD degradation lead to RF and FMN, respectively [20,22,23,24,25,26,37]. Specifically, a great number of consecutive reactions have been described and particularly catalyzed by essential enzymes [20,27,28,38,39]. Moreover, Reihl and Stolz [39] suggested that *S. cerevisiae* strains can increase the expression of the RF transporter in the condition of RF deficiency, and the uptake could depend on the growth rate, the sources of carbon and nitrogen, and the state of metabolism.

Our results showed that the release of RF and its derivative co-factors is strongly related to the growth conditions and the strains, revealing how the presence of RF in the environment can interfere with their synthesis and release. As expected, in the absence of RF and its derivatives in the cultural media, the activation of their biosynthetic pathway is required. The amount of RF excreted is stable at about 30–50 µg/L regardless of the strains and sampling time. On the other hand, probably depending on the metabolic state of the cells, the release of FMN and FAD differed in the three growth media; specifically, a higher amount of RF derivatives was released in the SMV medium with respect to SMV+RF and Chardonnay must. The release of the FDs was already important after 10 days of fermentation in all the strains, so we might speculate that the excretion starts at the early stationary phase; probably, this phenomenon is necessary to maintain the optimal intracellular content, starting when an intracellular over-accumulation of FDs occurs. According to Reihl and Stolz [39], *S. cerevisiae* has a mechanism to sense RF content, so we can hypothesize a negative feedback regulation-like caused by the perception of the high concentration of RF in the medium. In this context, a possible initial uptake by the membrane transporter Mch5p occurs, and when the internal concentration becomes enough and the optimal one, the cells start to excrete the over-accumulated products. Then, the intracellular and extracellular FDs amounts remained almost stable from the 10th and the 25th day, revealing a hypothetical lowering of their synthesis.

Cells show different behavior in Chardonnay must fermentations; actually, the RF content quantified in the extracellular environment was about 60 µg/L on the 10th day and significantly increased at approximately 100 µg/L on the 25th day, without apparent cell lysis. Conversely, FMN and FAD concentrations were significantly lower than those recorded in the synthetic grape musts. Cells in this condition prove to be able to keep the intracellular levels of the three compounds higher than that observed in the cells grown in the other two media. This phenomenon could be explained by the richness of nutrients in Chardonnay must, especially in terms of amino acid content. Indeed, the growth curves revealed a clear decrease in biomass after 10 days in the synthetic musts, while up to 25 days, the cell viability of all starters was still high in Chardonnay must. In addition, the addition of purines, serine, and threonine is usually used to promote the production of RF in industrial processes [40]. The presence of these compounds in must, besides that of proline that can induce the expression of the Mch5p transporter [41], could explain both the highest metabolic activity of the cells and the maintenance of FDs content.

Bearing in mind the biological interpretation, the total produced molecules of RF could explain the different metabolic activities of the four strains occurring in the three different media of growth. This estimation was considered since FAD/FMN turnover can assume a central role in cellular RF homeostasis by a FAD pyrophosphatase and an FMN phosphohydrolase, which respectively catalyze the following conversions: FAD + H_2_O→ FMN+AMP; FMN+H_2_O→ RF + Pi [42]. Particularly, in SMV, the highest production of RF molecules in all strains can be due to the activation of the RF metabolic route since this vitamin is necessary for FMN and FAD production and yeast survival. On the contrary, in SMV+RF, the lower amount of the produced RF can suggest that its high extracellular concentration acts in a negative feedback regulation. Finally, the results in Chardonnay must suggest that the found RF level can be considered optimal to limit the RF biosynthetic pathway and that together with an excess of nutrients in the medium of growth, the grape must can represent the best condition of growth, among the tested ones. Indeed, the amount of intracellular FDs in the extracts deriving from the growth in must suggests a more vigorous metabolic activity, considering the higher levels of the accumulated RF and FAD useful for several essential enzymatic reactions.

As far as the Met release, different trends can be described depending on the used growth medium. In the synthetic musts, a decrease in the strain biomasses indicates that the cell lysis led to an increase in the amino acid in the environment, while in Chardonnay must, the viable biomass remained almost constant between the 10th and 25th, and the strains probably consumed the available Met to survive.

The outcome of this work also showed different profiles of FDs accumulation in yeast strains, revealing that the tendency to release RF is both strain and medium formulation-dependent. Indeed, even if the AWRI796 strain secretes the highest amount of RF in all the tested conditions, all the strains revealed a lower release of RF during the fermentation in Chardonnay must than that observed in SMV and SMV+RF. Additionally, both the absence and the presence of high levels of RF in the environment seem to interfere with the biosynthesis of FDs.

Taking into account that the LST formation is avoided in a model spiked wine with either RF or Met at a concentration lower than 50 µg/L and 1.5 mg/L, respectively [9], our results suggest the potential risk of LST occurrence for the wines obtained from the fermentation of Chardonnay must with all the strains investigated. Indeed, the vinification trials showed that there is reliability and reproducibility of the laboratory trends in winery conditions, confirming that AWRI796 releases a higher amount of RF than EC1118 under the analyzed conditions. Moreover, as observed in the SMV medium, a limitation in bioavailable RF to yeasts could induce an activation of the biosynthetic pathway bringing to an overflow of the vitamin and, consequently, to its accumulation in the base wine. However, further investigations on how must composition can impact the final release of RF in finished wines and how the genetic determinants involved in the regulation of the Mch5p transporter of RF in yeasts work under oenological conditions will be necessary.

Lastly, in applicative terms, it will be useful to clarify in a relevant environment (winery level) what is the best strategy to limit the LST spoilage in either white or rosé, still and sparkling wines and if a combination of microbiological and technological approaches could be exploited in this context.

## Figures and Tables

**Figure 1 jof-09-00078-f001:**
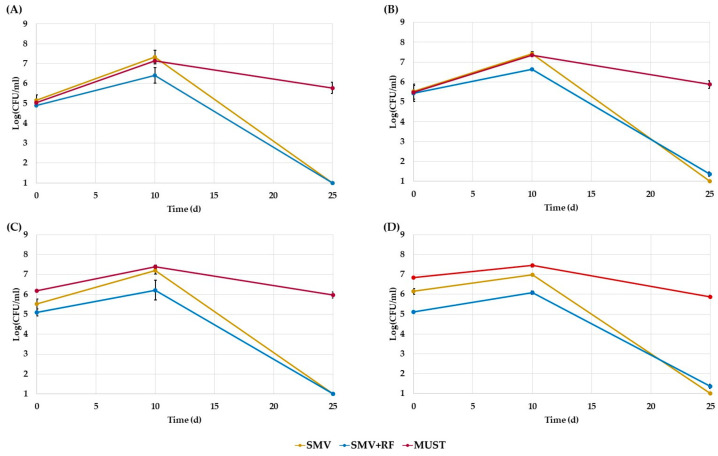
Trends of viable counts expressed as Log CFU/mL of the four strains in the three growth media (SMV = dark yellow line, SMV+RF = blue line, MUST = red line): (**A**) EC1118; (**B**) IOC18-2007; (**C**) LS2; (**D**) AWRI796. The results are reported as average data of three replicates, and the error bars indicate the standard deviations.

**Figure 2 jof-09-00078-f002:**
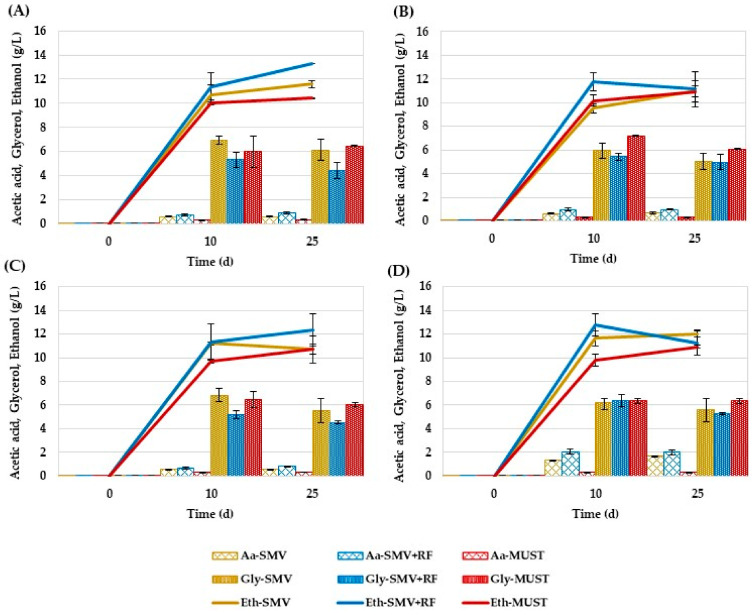
Amounts of acetic acid (Aa), glycerol (Gly) and ethanol (Eth) measured during the fermentation of the four strains. The dashed bars represent the production of acetic acid (Aa), while the full-colored ones the production of glycerol (Gly); the ethanol (Eth) production is represented by the continuous line. The results are reported as average data of three replicates and the error bars indicate the standard deviations. In each box are reported the behavior of a single strain in the three different media of growth (SMV = dark yellow, SMV+RF = blue line, MUST = red line), as follow: (**A**) EC1118; (**B**) IOC18-2007; (**C**) LS2; (**D**) AWRI 796.

**Figure 3 jof-09-00078-f003:**
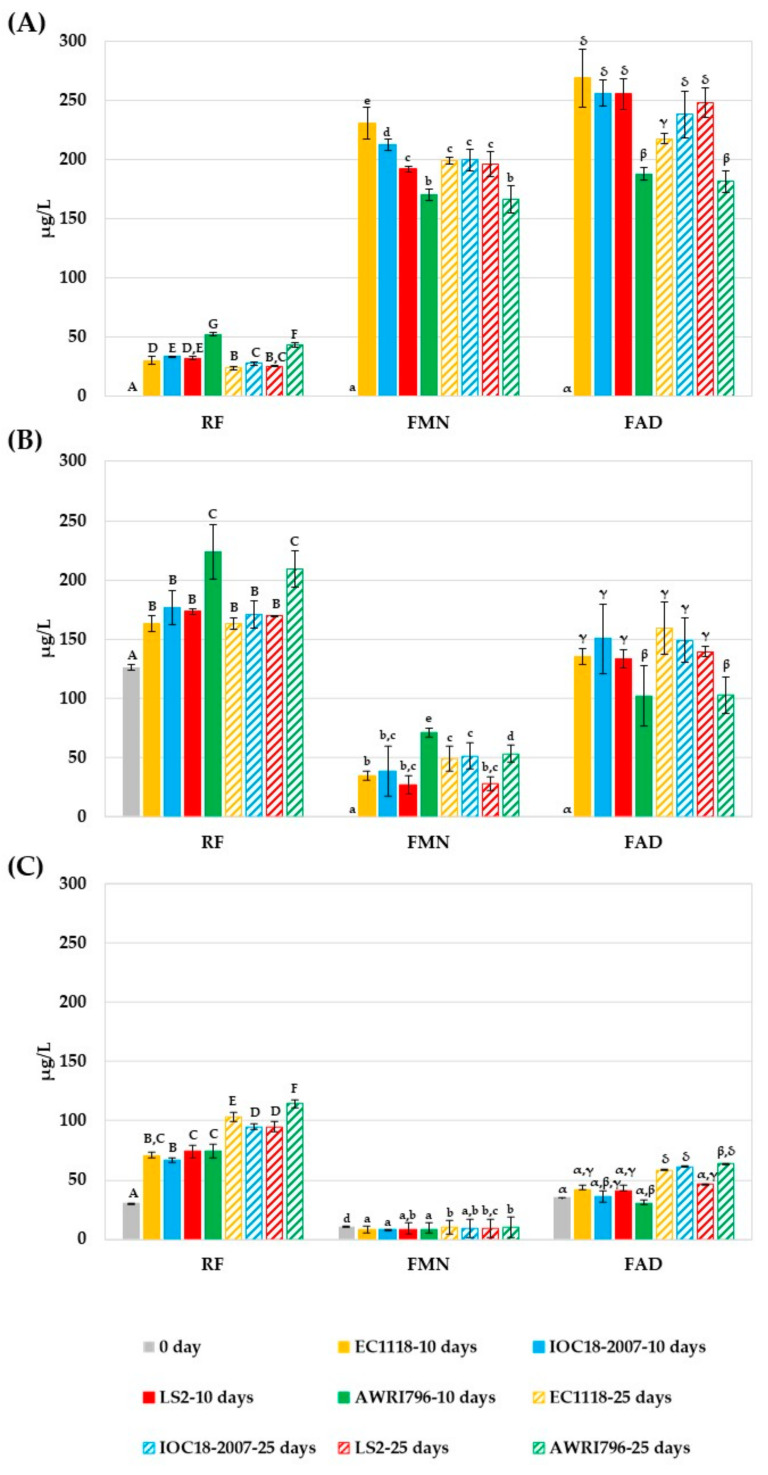
Extracellular FDs released during the fermentation processes: (**A**) SMV; (**B**) SMV+RF; (**C**) Must. Bars in grey describe the concentration at the starting point; the strains are represented by a specific color: EC1118 (yellow); IOC18-2007 (light-blue); LS2 (red); AWRI796 (green). The full-colored bars identify the concentrations after 10 days of fermentation, while the dashed ones show the concentrations after 25 days. The results are reported as average data of three replicates, and the error bars indicate the standard deviations. Capital, lowercase, and greek letters are used for RF, FMN, and FAD content, respectively. Different letters correspond to significant differences (*p* < 0.05).

**Figure 4 jof-09-00078-f004:**
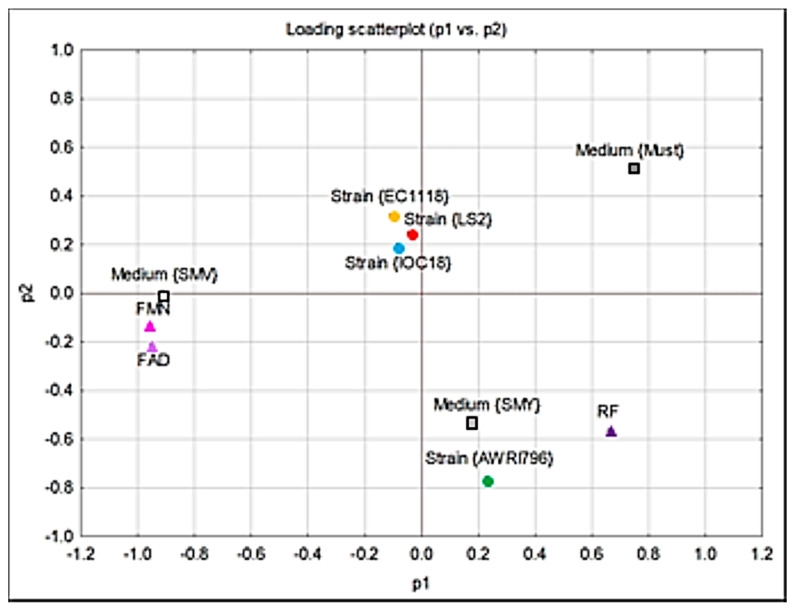
Scatter plot of the PCA analyses regarding the extracellular content of FDs.

**Figure 5 jof-09-00078-f005:**
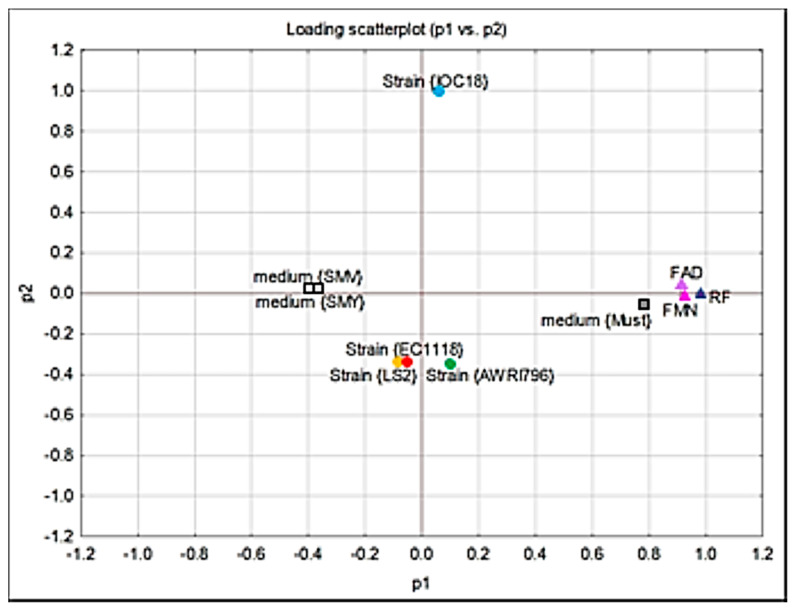
Scatter plot of the PCA analyses regarding the intracellular content of FDs.

**Figure 6 jof-09-00078-f006:**
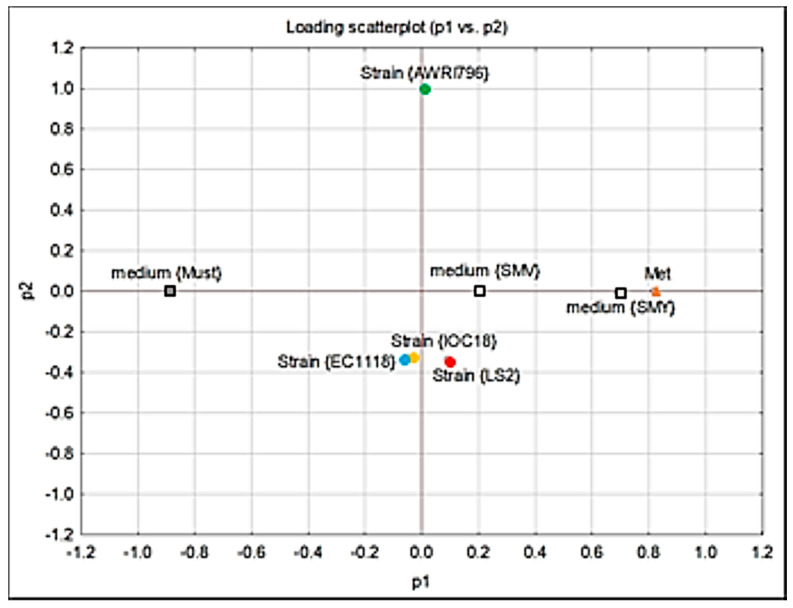
Scatter plot of the PCA analyses regarding the extracellular content of Met.

**Figure 7 jof-09-00078-f007:**
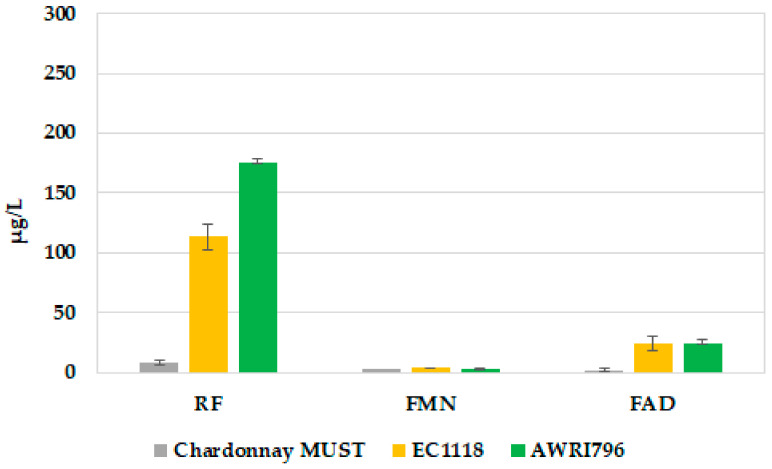
Extracellular FDs released at the end of the large-scale fermentation processes. Yellow bars indicate *S. cerevisiae* EC1118, and green bars *S. cerevisiae* AWRI796. Grey bars show the content of FDs in Chardonnay must. The results are reported as average data of three replicates, and the error bars indicate the standard deviations (*n* = 3).

**Table 1 jof-09-00078-t001:** Intracellular content of FDs during the yeast growths in the three cultural media. Reported data are mean values of three replicates with the standard deviation. Capital, lowercase, and greek letters are used for RF, FMN, and FAD content, respectively. Different letters correspond to significant differences (*p* < 0.05) among the strains grown in the same medium. nd = below the threshold limit of detection.

	Time (d)	EC1118	IOC18-2007	LS2	AWRI 796
RF (µg/L)	SMV				
	10	9.5 ± 3.9	10.6 ± 1.7	11.3 ± 6.3	10.2 ± 5.2
	25	6.3 ± 0.2	7.9 ± 2.9	7.5 ± 0.6	8.0 ± 1.5
	SMV+RF				
	10	12.9 ^A^ ± 0.3	12.9 ^A^± 1.4	13.7 ^AB^ ± 1.7	12.1 ^A^ ± 0.8
	25	13.2 ^AB^ ± 2.2	15.4 ^B^ ± 1.2	13.2 ^AB^ ± 0.1	17.1 ^C^ ± 0.7
	CHARDONNAY MUST				
	10	45.5 ^B^ ± 7.9	62.1 ^C^ ± 0.1	52.5 ^B^ ± 6.6	77.5 ^D^ ± 10.1
	25	15.2 ^A^ ± 1.1	22.0 ^A^ ± 0.1	17.5 ^A^ ± 0.4	18.6 ^A^ ± 1.5
FMN (µg/L)	SMV				
	10	nd	nd	nd	nd
	25	8.2 ± 3.8	nd	nd	nd
	SMV+RF				
	10	nd	nd	nd	nd
	25	nd	nd	nd	nd
	CHARDONNAY MUST				
	10	11.2 ^a^ ± 9.6	21.1 ^b^ ± 1.5	9.7 ^a^ ± 2.3	27.6 ^c^ ± 1.1
	25	nd	nd	nd	nd
FAD (µg/L)	SMV				
	10	21.3 ± 1.8	17.7 ± 2.4	23.9 ± 12.1	31.7 ± 0.1
	25	19.1 ± 3.3	14.4 ± 2.6	21.4 ± 2.3	nd
	SMV+RF				
	10	19.2 ^β^ ± 2.2	16.6 ^α^ ± 9.2	22.8 ^αβ^ ± 10.5	11.5 ^α^ ± 0.01
	25	9.2 ^α^ ± 0.1	18.2 ^αβ^ ± 0.0	nd	10.3 ^αβ^ ± 2.9
	CHARDONNAY MUST				
	10	83.0 ^βγ^ ± 29.8	114.3 ^γδ^ ± 1.3	81.5 ^β^ ± 43.3	131.5 ^δ^ ± 1.8
	25	nd	13.8 ^α^ ± 4.3	nd	11.3 ^α^ ± 0.01

**Table 2 jof-09-00078-t002:** Production of molecular RF (ng/10^6^ cells) during the fermentations in SMV, SMV+RF, and Chardonnay must. The results were divided into 10^13^ RF molecules in order to normalize the values.

	Time (d)	EC1118	IOC18-2007	LS2	AWRI 796
Potential production of RF (ng/10^6^ cells)	SMV				
	10	559	534	502	444
	25	468	485	491	430
	SMV+RF				
	10	207	254	212	331
	25	245	251	217	278
	CHARDONNAY MUST				
	10	75.9	66.9	80.2	80.4
	25	138	120	108.3	162

**Table 3 jof-09-00078-t003:** Extracellular release of Met during the fermentation processes in the three growth media. Reported data are mean values of three replicates with the standard deviation. Different letters correspond to significant differences (*p* < 0.05) among the strains grown in the same medium.

	Time (d)	EC1118	IOC18-2007	LS2	AWRI 796
Met (mg/L)	SMV				
	0	9.70 ^A^ ± 1.14			
	25	8.75 ^A^ ± 0.94	9.77 ^A^ ± 1.14	11.59 ^B^ ± 0.59	12.30 ^B^ ± 0.18
	SMV+RF				
	0	9.48 ^A^ ± 1.05			
	25	13.96 ^B^ ± 0.47	13.61 ^BC^ ± 1.41	14.43 ^B^ ± 1.44	12.23 ^C^ ± 1.32
	CHARDONNAY MUST				
	0	10.75 ^A^ ± 1.05			
	25	3.36 ^B^ ± 0.07	3.09 ^B^ ± 0.27	3.80 ^B^ ± 0.26	3.20 ^B^ ± 0.23

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
