# Peer review of "The Riboflavin Metabolism in Four Saccharomyces cerevisiae Wine Strains: Assessment in Oenological Condition and Potential Implications with the Light-Struck Taste"

_jof, 2023, doi:10.3390/jof9010078_

Round 1

Reviewer 1 Report (Previous Reviewer 2)

All comments have been successfully incorporated.

Reviewer 2 Report (Previous Reviewer 1)

All my questions have been answered in a satisfactory way and your work has improved substantially.

I recommend that your article be published now.

This manuscript is a resubmission of an earlier submission. The following is a list of the peer review reports and author responses from that submission.

Round 1

Reviewer 1 Report

SOME RECOMMENDATIONS FOR THE AUTHORS.

Title: The title must be changed. This work is focused to study the metabolism of Riboflavin during the alcoholic fermentation of a Chardonnays grape must to obtain a still wine, which can be used, or not, for sparkling wine elaboration by carrying out a second fermentation.

I suggest to delete the word sparkling in title.

 Abstract

Line 27. Flavones are yellow pigments belonging to a subclass of polyphenolic compounds. Apigenin, Luteolin, Baicalein, Chrysin, are some examples of them. Authors should change the Flavone word in the complete manuscript to avoid confusions with the aim of this study, the Riboflavin and its related co-factors FM and FAD..

Introduction

Line 68 : Please change the normal letter of S. cerevisiae to italic letters.

Lines 68-70. This sentence should be rewritten. It is not easy to understand.

Lines 90-92. Please , explain in a west way this sentence: At this purpose, both intra- and extracellular production of flavones, by four commercial yeasts, grew in synthetic grape must medium (containing 0 or about 125 μg/L of RF) and Chardonnay must was monitored during the AF.

Flavones are secondary plant metabolites and only are described its production in mutants S. cerevisiae yeast. Why are important in this work?

Material and methods

Lines 112-114. Please, add the composition of Chardonnay grape must (pH, reducing sugars, tartaric acid, malic acid, etc,) It is well known that the osmotic stress due to the solved solutes affect the yeast metabolism.

Lines 113-114. · The added disodium metabisulfite (30 mg/L) is equivalent to a dose of 20.2 mg/L in SO2 (molecular). Authors should clarify how Authors should clarify how they express the final content (as SO2 or as Na2S205).  

Lines 123-124. Please establish an equivalent range in cells/mL for the yeast’s population, in view to a most easy comparison with other studies.

Line 140. Specify the volume.

Line 169. Authors say:  Briefly, flavones, namely RF, FAD and FMN….

Are the authors sure that RF, FAD and FMN are flavone compounds?  RF, FAD and FMN are not flavin compounds? 

Did  the authors confuse the Flavin derivatives with the flavone compounds?

Flavones and other flavonoid compounds in the wine chemistry are polyphenolic compounds with an oxygen heterocycle in their molecular structure, while the flavin derivatives have a nitrogen heterocycle.

Riboflavin (vitamin B2), is the precursor of flavin cofactors as FAD and FMN.

Flavonoids are synthetized by plants or genetically modified Saccharomyces yeasts.

The authors should take in account these comments and review carefully the manuscript before it was accepted for publication.

There are 41 citations in text of the flavone word and they should be revised carefully according to their context.

 Line 177: Specify the range values for R2 to the calibration curves.

Results

Please, before anything authors should check if they mean flavin derivatives or flavone derivatives.

Line 388. In the PCA plot performed on the extracellular determination of flavones the second cluster is composed exclusively of AWRI796. However, the variance explained in these first two components summarizes only 54 % of the total variance. Have you tried plotting other principal components? Please, explain this aspect..

 Discussion

 Please review the use of the word Flavone and change it, if necessary.

 Tables and Figures.

Table 1. Legend: RF, FMN and FAD are not flavones. They are Flavin-derivatives. Please, correct it.

Descriptions of tables and figures should be improved, especially abbreviations should carry their meaning.

  References.

All references listed are cited in text.

Author Response

Reviewer 1

Comments and Suggestions for Authors

Title: The title must be changed. This work is focused to study the metabolism of Riboflavin during the alcoholic fermentation of a Chardonnays grape must to obtain a still wine, which can be used, or not, for sparkling wine elaboration by carrying out a second fermentation.

I suggest to delete the word sparkling in title.

Thank you for your suggestion. We changed the title of the paper.

Abstract

Line 27. Flavones are yellow pigments belonging to a subclass of polyphenolic compounds. Apigenin, Luteolin, Baicalein, Chrysin, are some examples of them. Authors should change the Flavone word in the complete manuscript to avoid confusions with the aim of this study, the Riboflavin and its related co-factors FM and FAD.

Done. We changed the “Flavone” word in the whole manuscript.

Introduction

Line 68 : Please change the normal letter of S. cerevisiae to italic letters.

Done.

Lines 68-70. This sentence should be rewritten. It is not easy to understand.

Thank you. We changed the sentence “After all, the generation of S. cerevisiae deletion mutants of the genes involved in RF bio-synthetic pathway (RIB1, RIB2, RIB3, RIB4, RIB5, RIB7) allowed to obtain auxotrophic strains, suggesting the essential nature of flavones.” as follow: “After all, molecular experiments to obtain S. cerevisiae deletion mutants of the genes involved in RF biosynthetic pathway (RIB1, RIB2, RIB3, RIB4, RIB5, RIB7) has been reported; these studies demonstrated the essential nature of the FDs because the generated mutants became non-viable”

Lines 90-92. Please, explain in a west way this sentence: At this purpose, both intra- and extracellular production of flavones, by four commercial yeasts, grew in synthetic grape must medium (containing 0 or about 125 μg/L of RF) and Chardonnay must was monitored during the AF.

In order to clarify the meaning, we changed the sentence as follow: “At this purpose, the intra- and extracellular contents of RF and its related co-factors, were monitored during the AF of four commercial starter yeasts growing in synthetic media (similar in composition to a grape must and containing 0 or about 125 µg/L of RF) and Chardonnay must.”

Flavones are secondary plant metabolites and only are described its production in mutants S. cerevisiae yeast. Why are important in this work?

Thank you for your comment. We changed the “Flavone” word in the whole manuscript.

Material and methods

Lines 112-114. Please, add the composition of Chardonnay grape must (pH, reducing sugars, tartaric acid, malic acid, etc,) It is well known that the osmotic stress due to the solved solutes affect the yeast metabolism.

Thank you for your comment. We added the following information: “”sugars 240 g/L, tartaric acid 7 g/L, pH = 3.5”.

Lines 113-114. · The added disodium metabisulfite (30 mg/L) is equivalent to a dose of 20.2 mg/L in SO2 (molecular). Authors should clarify how Authors should clarify how they express the final content (as SO2 or as Na2S205).  

We express the final content as sodium metabisulfite.

Lines 123-124. Please establish an equivalent range in cells/mL for the yeast’s population, in view to a most easy comparison with other studies.

We modify the optical density values with the equivalent range in cells/mL “at the concentration of 1-5 x 106 CFU/mL”.

Line 140. Specify the volume.

We cannot specify the volume, because it depended on the specific growth of the strains measured by optical density values.

Line 169. Authors say:  Briefly, flavones, namely RF, FAD and FMN….

Are the authors sure that RF, FAD and FMN are flavone compounds?  RF, FAD and FMN are not flavin compounds? 

Did  the authors confuse the Flavin derivatives with the flavone compounds?

Flavones and other flavonoid compounds in the wine chemistry are polyphenolic compounds with an oxygen heterocycle in their molecular structure, while the flavin derivatives have a nitrogen heterocycle.

Riboflavin (vitamin B2), is the precursor of flavin cofactors as FAD and FMN.

Flavonoids are synthetized by plants or genetically modified Saccharomyces yeasts.

The authors should take in account these comments and review carefully the manuscript before it was accepted for publication.

There are 41 citations in text of the flavone word and they should be revised carefully according to their context.

We agree. We changed the “Flavone” word in the whole manuscript.

Line 177: Specify the range values for R2 to the calibration curves.

The range was 0.997-0.999. We added this information as suggested.

Results

Please, before anything authors should check if they mean flavin derivatives or flavone derivatives.

Thank you. We changed the “Flavone” word in the whole manuscript.

Line 388. In the PCA plot performed on the extracellular determination of flavones the second cluster is composed exclusively of AWRI796. However, the variance explained in these first two components summarizes only 54 % of the total variance. Have you tried plotting other principal components? Please, explain this aspect.

We plotted only the first two components because the results indicated that the first two components were significant to describe a clear relationship between the analysed parameters.

 Discussion

 Please review the use of the word Flavone and change it, if necessary.

Thank you. We changed the “Flavone” word in the whole manuscript.

Tables and Figures.

Table 1. Legend: RF, FMN and FAD are not flavones. They are Flavin-derivatives. Please, correct it.

Done.

Descriptions of tables and figures should be improved, especially abbreviations should carry their meaning.

Done.

References.

All references listed are cited in text.

Thank you.

Reviewer 2 Report

Is the manuscript clear, relevant for the field and presented in a well-structured manner?

Is the manuscript scientifically sound and is the experimental design appropriate to test the hypothesis?

Please change the title – you have only tested four commercial strains in an artificial medium in 250ml flask…. You must repeat a minimum volume of 25L three times to make a statement for the wine industry

Are the manuscript’s results reproducible based on the details given in the methods section?

This publication is only a description of the metabolism in four different commercial wine yeast in an artificial must. The experiments were carried out in 250ml flask. Due to this size, a relevant statement for wine practice is not possible. Therefore, it is only a potential analysis for the light-struck taste production in these yeasts.

Are the cited references mostly recent publications (within the last 5 years) and relevant? Does it include an excessive number of self-citations?

Yes, it is ok.

Are the figures/tables/images/schemes appropriate? Do they properly show the data? Are they easy to interpret and understand? Is the data interpreted appropriately and consistently throughout the manuscript? Please include details regarding the statistical analysis or data acquired from specific databases.

Many legends are missing eg..Figure 1, 2  SMV , RF….Eth, Gly

Please make it easier to interprete.

Are the conclusions consistent with the evidence and arguments presented?

Please change the content.

Line 41 – co- authored…

L150-154 What was the purpose of this analysis?

L705-L711 As a practitioner, I cannot agree with that. The sample amount is too small.

Author Response

Reviewer 2

Comments and Suggestions for Authors

Is the manuscript scientifically sound and is the experimental design appropriate to test the hypothesis?

Please change the title – you have only tested four commercial strains in an artificial medium in 250ml flask…. You must repeat a minimum volume of 25L three times to make a statement for the wine industry

Thank you for your suggestion. We changed the title of the manuscript.

This work is part of the “Enofotoshield '' project that aims at reducing the LST defect in white and sparkling wines. The manuscript presents the RF metabolism, including its FMN and FAD co-factors, under oenological condition at laboratory scale. The results will be useful for the selection of yeasts to be used in 50 L pilot AF and secondary in-bottle fermentations in a relevant environment (cellar).

Are the manuscript’s results reproducible based on the details given in the methods section?

This publication is only a description of the metabolism in four different commercial wine yeast in an artificial must. The experiments were carried out in 250ml flask. Due to this size, a relevant statement for wine practice is not possible. Therefore, it is only a potential analysis for the light-struck taste production in these yeasts.

We agree., This work provides a description of the RF, including its FMN and FAD co-factors, metabolism in four different commercial wine yeasts at laboratory scale. Thus, we modify the title accordingly. The further scale up at 50 L will allow the investigation of the occurrence of the LST defect in a more realistic scenario, moving from TRL4 to TRL5.

Are the cited references mostly recent publications (within the last 5 years) and relevant? Does it include an excessive number of self-citations?

Yes, it is ok.

Thank you.

Are the figures/tables/images/schemes appropriate? Do they properly show the data? Are they easy to interpret and understand? Is the data interpreted appropriately and consistently throughout the manuscript? Please include details regarding the statistical analysis or data acquired from specific databases.

Many legends are missing eg..Figure 1, 2  SMV , RF….Eth, Gly

Please make it easier to interprete.

The graphs and legends have been improved.

Moreover, we decided to change the name of the medium “SMVRF” IN “SMV+RF”, to clarify that the two laboratory media only differ each other for the presence of RF.

Are the conclusions consistent with the evidence and arguments presented?

Please change the content.

We changed the discussion as follow:

“Since yeast cell life and resilience to stress factors can be strongly influenced by nutrient availability, clarifying which are the growth conditions that interfere with the metabolism of RF can help to limit its release during grape must fermentation. In this work the effect of exogenous RF on the FDs biosynthesis has been evaluated in synthetic media and real Chardonnay must; particularly, we were going to determine if extracellular RF content might serve as a signal to report a sufficient vitamin availability to cells. Since yeast metabolism can be strongly influenced by nutrient availability, clarifying which are the growth conditions that can interfere on the metabolism of RF can help to limit its release during grape must fermentation. Moreover, the outcome of the study could improve the required knowledge for the setup of suitable approaches to counteract RF accumulation in wine, taking the form of a cure and not a remedy for the LST defect. For this purpose, the used cultural media were formulated in order to support the yeast growth in the tumultuous fermentative phase in terms of nutrients, except for the presence of RF: SMV was lacking in the vitamin content, while SMV+RF contained about 125 µg/L RF. The experiments in Chardonnay must were included in the experimental plan, though RF concentration in this case was usually too low for triggering LST [11]. Therefore, the conditions present in SMV medium can be considered more similar to those of the must.

Plate counts confirmed that the viability remained stable on 10th day (primary fermentation phase) and then it started to slightly decline [36,37]. Moreover, the analysis of the fermentative metabolites allowed to confirm that in all the tested conditions the four selected strains were able to complete a simulated fermentation process, confirming the results described by Zhang and co-authors [38].

The biosynthetic pathways of RF and its derivative coenzymes have been deeply investigated in S. cerevisiae; they are strictly related since FMN and FAD degradation lead to RF and FMN, respectively [21,23,24-27,39]. Specifically, a great number of consecutive reactions have been described and particularly catalyzed by essential enzymes [21,28-30, 40]. Moreover, Reihl and Stolz [40] suggested that S. cerevisiae strains can increase the expression of the RF transporter in condition of RF deficiency and the uptake could depend on the growth rate, the sources of carbon and nitrogen, and the state of metabolism.  

Our results showed that the release of RF and its derivative co-factors is strongly related to the growth conditions and the strains, revealing how the presence of RF in the environment can interfere with their synthesis and release. As expected, in the absence of RF and its derivatives in the cultural media, the activation of their biosynthetic pathway is required. The amount of RF excreted is stable at about 30-50 µg/L regardless of the strains and sampling time. On the other hands, probably depending on the metabolic state of the cells, the release of FMN and FAD differed in the three growth media; specifically, higher amount of RF derivatives were released in SMV medium with respect to SMV+RF and Chardonnay must. The release of the FDs was already important after 10 days of fermentation in all the strains, so we might speculate that the excretion starts at the early stationary phase; probably, this phenomenon is necessary to maintain the op-timal intracellular content, starting when an intracellular over accumulation of FDs occurs. According to Reihl and Stolz [40] S. cerevisiae has a mechanism to sense RF content, so we can hypothesize a negative feedback regulation-like caused by the perception of the high concentration of RF in the medium. In this context, a possible initial uptake by the membrane transporter Mch5p occurs and when the internal concentration becomes enough and the optimal one, the cells start to excrete the over accumulated products. Then, the intracellular and extracellular FDs amounts remained almost stable from the 10th and the 25th day, revealing a hypothetical lowering of their synthesis.

Cells show a different behavior in Chardonnay must fermentations; actually, the RF content quantified in the extracellular environment was about 60 µg/L on the 10th day and significantly increased at approximately 100 µg/L on the 25th day, without apparent cell lysis. Conversely, FMN and FAD concentrations were significantly lower than those recorded in the synthetic grape musts. Cells in this condition prove to be able to keep the intracellular levels of the three compounds higher than that observed in the cells grown in the other two media. This phenomenon could be explained by the richness of nutrients in Chardonnay must, especially in terms of amino acidic content. Indeed, the growth curves revealed a clear decrease of biomass after 10 days in the synthetic musts, while up to 25 days the cell viability of all starters was still high in Chardonnay must. Also, the addition of purines, serine and threonine are usually used to promote the production of RF in industrial processes [42]. The presence of these compounds in must, beside that of proline that can induce the expression of the Mch5p transporter [43], could explain both the highest metabolic activity of the cells and the maintenance of FDs content.

Bearing in mind the biological interpretation, the total produced molecules of RF could explain the different metabolic activity of the four strains occurring in the three different media of growth. This estimation was considered since FAD/FMN turnover can assume a central role in cellular RF homeostasis, by a FAD pyrophosphatase and a FMN phosphohydrolase, which respectively catalyze the following conversions: FAD + H2O à FMN+AMP; FMN+H2O à RF + Pi [43]. Particularly, in SMV the highest production of RF molecules in all strains can be due to the activation of the RF metabolic route since this vitamin is necessary for FMN and FAD production and the yeast survival. On the contrary, in SMV+RF the lower amount of the produced RF can suggest that its high extracellular concentration acts in a negative feedback regulation. Finally, the results in Chardonnay must suggest that the found RF level can be considered optimal to limit the RF biosynthetic pathway, and that together with an excess of nutrients in the medium of growth, the grape must can represent the best condition of growth for the strains. In-deed, the amount of intracellular FDs in the extracts deriving from the growth in must suggested a more vigorous metabolic activity, considering the higher levels of the ac-cumulated RF and FAD useful for several essential enzymatic reactions

As far the Met release, different trends can be described depending on the used growth medium. In the synthetic musts a decrease of the strain biomasses indicates that the cell lysis led to an increase of the amino acid in the environment, while in Chardonnay must the viable biomass remained almost constant between the 10th and 25th and the strains probably consumed the available Met to survive.

However, taking into account that in a model spiked wine with either RF and Met the LST formation is avoided at lower concentration of the two compounds at 50 µg/L and 1.5 mg/L, respectively, our results indicate that a potential production of volatile sulfur compounds due to LST could also derive from the proliferation in Chardonnay grape must of all the four investigated strains.

The outcome of this work also shown different profiles of FDs accumulation in yeast strains, revealing that the tendency to release RF into the medium is also strain de-pendent and thus indicate that the strategy to select a low RF-producer yeast can be pursued to limit LST in wines and sparkling wine. Indeed, AWRI796 strain secretes the highest amount of RF in all the tested conditions and therefore it is unsuitable to deal with the defect in the wine.

Additionally, both the absence and the presence of high level of RF in the environment seem to interfere with the biosynthesis of FDs. Indeed, the presence of low amounts of both RF and Met, as occurred for Chardonnay must, appears to be the most promising condition to limit their high release [9] (Figure 7).

Since RF uptake operates by a facilitated diffusion, further studies addressed to the study of the genetic determinants involved in the regulation of the Mch5p transporter and their expression in different environmental conditions will be necessary. Moreover, it will be useful to clarify in a relevant environment (winery level) what is the best strategy to limit the LST spoilage in either white or rosé, still and sparkling wines and if a combination of microbiological and technological approaches could be exploited in this context.”

Line 41 – co- authored…?

This sentence has been cancelled.

L150-154 What was the purpose of this analysis?

This analysis was performed to verify the efficiency of the lysis protocol in terms of “number of unbroken cells/number of total cells) x 100”. In general, less than 3% of cells remained intact.

L705-L711 As a practitioner, I cannot agree with that. The sample amount is too small.

We agree. We modified the sentence as follow: “Additionally, both the absence and the presence of high level of RF in the environment seem to interfere with the biosynthesis of FDs. Indeed, the presence of low amounts of both RF and Met, as occurred for Chardonnay must, appears to be the most promising condition to limit their high release [9] (Figure 7).”

Reviewer 3 Report

The work entitles “Assessment of the riboflavin metabolism in Saccharomyces cerevisiae oenological strains as part of the light-struck taste occurrence in sparkling wine production” showed the comparison between different mediums and strain in order to avoid the low quality of wine as consequence of the riboflavin and methionine transformation.

The topic is interesting but in my opinion the most important lack in the work is that the authors does not make an experiment that associated the best medium and strain with the reduction of flavor defect. It is true that indirectly the riboflavin and met concentration can reduce this effect, I am not sure if in the real wine processing the use of a specific strain and medium avoid the flavor affectation under light. I suggested that the authors should make an experiment using the best condition under real wine production parameter and intentionality expose the wine under light and corroborate that the affection is avoiding by the use of the strain that they proposed. I understand that wine production takes even years, but some preliminary fermentation avoiding the aging could be performant, even the author suggest that this part is necessary at the end of the discussion.

Also I have the next observations:         

·         the connection of introduction and results section is not clear. Figure 2 showed information such as glycerol or acetic acid that is not mentioned in the introduction, if measure this parameter is important, why is not mentioned in introduction? In my opinion the format of the graphics is not clear since the color and the texture of the graphs is too similar. Also the nomenclature is not easy, for example in figure 2, aa-SMV and aa-SMVRF is almost the same, I have to back to the beginning of the paper to remember the nomenclature.

·         I am not sure about the utility of figure 1 presented as this form, no difference was found in each strain just in the culture medium, I think that is easer made 1 graph comparing the medium in state of strains.

·         The methionine origin is intracellular as consequence of the metabolism of the yeast? So in the fermentations condition the yeast has to export met? I am not sure if the quantity of met exported by the yeast is enough to produce the effect on the flavor or is necessary some external source of met.

·         In my opinion the 3 fists paragraphs are unnecessary, since your discussion must be focus on riboflavin and met relationship, as the same as the introduction, if are important the resistance of the yeast, the glycerol concentration, etc. you must be mentioned even in the title of the work.

Author Response

Reviewer 3

Comments and Suggestions for Authors

The work entitles “Assessment of the riboflavin metabolism in Saccharomyces cerevisiae oenological strains as part of the light-struck taste occurrence in sparkling wine production” showed the comparison between different mediums and strain in order to avoid the low quality of wine as consequence of the riboflavin and methionine transformation.

The topic is interesting but in my opinion the most important lack in the work is that the authors does not make an experiment that associated the best medium and strain with the reduction of flavor defect. It is true that indirectly the riboflavin and met concentration can reduce this effect, I am not sure if in the real wine processing the use of a specific strain and medium avoid the flavor affectation under light. I suggested that the authors should make an experiment using the best condition under real wine production parameter and intentionality expose the wine under light and corroborate that the affection is avoiding by the use of the strain that they proposed. I understand that wine production takes even years, but some preliminary fermentation avoiding the aging could be performant, even the author suggest that this part is necessary at the end of the discussion.

Thank you for your suggestion.

The study is part of the “Enofotoshield '' project that aims at reducing the LST defect in white and sparkling wines. The manuscript presents the RF metabolism, including its FMN and FAD co-factors, under oenological condition at laboratory scale. The results will be useful for the selection of yeasts to be used in 50 L pilot AF and secondary in-bottle fermentations in a relevant environment (cellar).

Also I have the next observations:        

  • the connection of introduction and results section is not clear. Figure 2 showed information such as glycerol or acetic acid that is not mentioned in the introduction, if measure this parameter is important, why is not mentioned in introduction? In my opinion the format of the graphics is not clear since the color and the texture of the graphs is too similar. Also the nomenclature is not easy, for example in figure 2, aa-SMV and aa-SMVRF is almost the same, I have to back to the beginning of the paper to remember the nomenclature.

Thank you for your advices.

  • We decided to follow the fermentation processes, evaluating the three fermentative parameters and the sugar consumption, in order to perform a comparison among the conditions of growth. The comparison showed that the strains were in optimal conditions to perform a fermentation of 25 days.
  • We changed the format of the graphs, as suggested, in order to show the results in a clearer manner; moreover, we modified the legends inserting the explanation of the nomenclature, to simplify the reading of the results.

  • I am not sure about the utility of figure 1 presented as this form, no difference was found in each strain just in the culture medium, I think that is easer made 1 graph comparing the medium in state of strains.

Figure 1 has been modified as suggested.

  • The methionine origin is intracellular as consequence of the metabolism of the yeast? So in the fermentations condition the yeast has to export met? I am not sure if the quantity of met exported by the yeast is enough to produce the effect on the flavor or is necessary some external source of met.

The amount of Met depends on both the yeast metabolism and the composition of the medium of growth. During the growth in must the yeast strain were able to metabolize the external Met, indeed the final concentration was lower than that recorded at the initial time. However, to the final concentration of Met contribute the lysis of the cells, so the effect on the flavour depends on both to the intra- and extracellular content of Met.

  • In my opinion the 3 fists paragraphs are unnecessary, since your discussion must be focus on riboflavin and met relationship, as the same as the introduction, if are important the resistance of the yeast, the glycerol concentration, etc. you must be mentioned even in the title of the work.

We modified the discussion accordingly; since the fermentative parameters have been evaluated to compare the growth of the strains in the three cultural media, comments on the sugar/glycerol/acetic acid concentrations have been limited to the scope for which they have been assessed.

Round 2

Reviewer 3 Report

I maintain my opinion that carrying out an experiment where it is verified that the problem of flavor in the wine is avoided when using the strain and the culture medium suggested by the authors is mandatory for the acceptance of the manuscript, since that was the scope, focus and context that the authors decided to give the article. The authors would have to modify the approach to avoid this omission, however, I understand that it is not that simple to do, so I leave it to the opinion of the editor.

Regarding my other suggestions, the authors improved the article and attended to all my observations.